PREPARED FOR SUBMISSION TO JHEP

SLAC-PUB-17680

# Designing Observables for Quantum Interference in Jets at Subleading Color

**Andrew J. Larkoski**

*SLAC National Accelerator Laboratory, 2575 Sand Hill Road, Menlo Park, CA 94025, USA*

*E-mail:* larkoski@slac.stanford.edu

ABSTRACT: The large-$N_c$ or topologically planar limit of gauge theories can be considered as a classical limit because all gauge bosons are distinguishable particles and therefore cannot exhibit interference. Quantum effects due to the flow of color therefore arise starting at subleading in $1/N_c$. We introduce kinematic observables explicitly sensitive to effects at subleading color formed from the ratio of interfering to squared color-ordered amplitudes. Such observables are in general not infrared and collinear safe, so we introduce angular observables defined from appropriate multi-point energy correlators motivated by the form of color-ordered amplitudes. We demonstrate that color interference effects are manifest as sinusoidal oscillation in the simplest system, a collinear jet with three particles, and show the limitations of predicting this observable in all-purpose, leading-color parton shower Monte Carlos.

## 1   Introduction

Signatures of quantum mechanics can be subtle and challenging to measure in the busy environment of a particle collision experiment. Interference of orthogonal states, for example, necessarily occurs whenever the constraints imposed by measurements are consistent with multiple histories, as observed in the double slit experiment. An interference pattern is observed on the screen as long as the measurements performed are consistent with identical photons traveling through either slit. Isolation of definite interfering states at a collider is obscured by the huge numbers of particles produced, substantial backgrounds, and the way in which measurements are performed, typically only sensitive to particle momenta.

Nevertheless, some efforts have succeeded in defining observables that are directly sensitive to interference of intermediate states. One recent example from Ref. [1] is sensitive to the interference of the two helicity states of an on-shell gluon. The distribution of this observable exhibits a $\cos(2\phi)$ pattern, a consequence of the fact that the difference between the two spin states is 2. The procedure used in Ref. [1] can be generalized to construct other observables sensitive to quantum interference. Two features are specifically exploited and we will study in more detail in this paper.

First, states in a quantum system are labeled by their conserved quantum numbers, like a massless particle's helicity, and so the interfering intermediate states must carry some distinct quantum numbers. This observation is highly restricting because the states of the Standard Model are its short-distance particle content and they have very few quantum numbers. Just considering the QCD sector, the only relevant, non-trivial quantum numbers

shared by all quarks and gluons are spin and color. As the interference of spin states has been studied, we will focus on the interference of intermediate color states. Second, backgrounds and contamination can be significantly reduced by exploiting the collinear divergences of perturbative QCD. In the collinear limit, contamination effects are suppressed by the area of the small angular region, while "signal", physical quarks and gluons, are unsuppressed, regardless of the size of the angular region. Therefore, we will work to construct observables sensitive to the interference of color states in high-energy QCD jets.

We will do this by considering the SU(3) gauge symmetry of QCD under which quarks and gluons are charged to lie on a spectrum of general SU($N_c$) gauge theories. The large and small $N_c$ limits have interesting features that we can exploit. It is well known that in the large-$N_c$ limit, an SU($N_c$) gauge theory reduces to a planar theory in which interactions are exclusively described by diagrams that can be drawn on a plane or the Poincaré sphere [2]. Large-$N_c$ means that all gluons necessarily carry distinct color quantum numbers, and are therefore distinguishable particles. Thus, the large-$N_c$ limit can also be thought of as a classical limit in which no quantum interference occurs [3]. At the other end of the $N_c$ spectrum, as $N_c \to 1$, the gauge theory reduces to quantum electrodynamics (QED). Photons in QED are all identical, indistinguishable particles because all photons carry no color, and therefore have a unique color quantum number. Thus, as well-familiar from the double slit experiment, identical photons exhibit quantum interference. The fact that QCD is an SU(3) gauge theory of color means that gluons are partly distinguishable and partly identical and this feature makes observing quantum interference due to color effects especially interesting.

With this background, we would like to construct an observable that is only non-zero if there is quantum interference due to multiple intermediate color states contributing to a measurement. Again, restricting to quantities measured at a collider, we consider kinematic observables that are exclusively dependent on particle momenta. Because color interference only exists at finite-$N_c$, this observable will further only be non-trivial beyond leading order in the large-$N_c$ approximation. Thus, it can be a powerful tool for demonstrating that a parton shower correctly includes subleading color effects, and beyond leading color or even full color showers are a very active area of contemporary research [4–17].

Our procedure for constructing this observable is the following. We think of the identification of subleading color or its interference as a binary discrimination problem [18]. Our goal will be to discriminate subleading color, the signal, from leading color, the background, physics. As a binary discrimination problem, we immediately know the optimal observable for discrimination by the Neyman-Pearson lemma [19]: the likelihood ratio. The likelihood ratio is just the ratio of the relevant signal to background probability distributions. As we restrict measurements to particle momenta, probability distributions live on relativistic phase space and are therefore absolute squared scattering amplitudes. The observable we consider is therefore the ratio of the squared amplitude for the production of gluons in a finite-$N_c$ gauge theory to the squared amplitude in the leading large-$N_c$ limit. This ratio can therefore be appropriately defined from the basis of color-ordered amplitudes [20–24].

Further, to reduce background by exploiting the collinear divergences of theories like

QCD, we will focus on color interference in narrow jets. In the limit that the interfering gluons become collinear, the ratio of squared amplitudes defining the observable simplifies to a ratio of collinear splitting functions. Despite this simplification, ratios of perturbative amplitudes are not in general infrared and collinear (IRC) safe and therefore this observable cannot in general be calculated in perturbation theory. However, by considering simplified collinear amplitudes with particular particle helicity configurations, we demonstrate that the subleading color observable is exclusively dependent on relative angles between pairs of particles, essentially taking the form of the jet color ring introduced in Ref. [25]. A functional form that is exclusively dependent on relative particle angles can be uplifted to an IRC safe observable by embedding it into multi-point energy correlators [26], generalizations of the energy-energy correlator [27], an early observable for studying jet production in electron-positron collisions.

The color interference pattern is encoded in the relative angles between gluons about the mother particle that emitted them. In a realistic experiment, of course only color-neutral hadrons are observable, and so there is significant ambiguity as to which particle can be identified as the initiator. We solve this problem practically by focusing on heavy-flavor jets, in which a bottom quark, for example, was produced in the short-distance collision and then emitted gluons at longer distances. The momentum direction of long-lived bottom hadrons is well-reproduced from displaced vertices in a tracking system [28, 29] and provides a concrete and unambiguous axis about which to measure correlations.

The outline of this paper is as follows. In Sec. 2, we review the color-ordered amplitude formalism and construct the likelihood ratio between finite- and large-$N_c$ squared amplitudes. Most calculational details are restricted to the first non-trivial case corresponding to the process $e^+e^- \to qgg\bar{q}$. In Sec. 3, we present the embedding of this amplitude ratio into IRC safe energy correlators, defined as a function of the relative angles between the measurement probes in the correlator. We also present calculations for the interference pattern at subleading color here, using the $1 \to 3$ collinear splitting functions. Comparison with parton shower Monte Carlo is presented in Sec. 4. This comparison is very limited here as we only present results from a leading-color parton shower, in hopes that this work inspires groups working on accuracy at subleading color to test on their parton showers. We conclude in Sec. 5, and look forward to measurements of subleading color effects on collider data.

## 2 An Observable for Color Interference

A scattering amplitude $\mathcal{A}$ in a non-Abelian gauge theory like QCD can always be expanded in terms of color-ordered basis amplitudes $A_n$ multiplied by an appropriate color matrix product $\mathbb{T}_n$ [20–24]:

$$\mathcal{A} = \sum_{n \text{ color orders}} \mathbb{T}_n A_n \,. \tag{2.1}$$

Here, $n$ represents topologically-distinct particle orderings in the amplitude, according to the color representation that they carry. Color-ordered amplitudes assume that gluons are

non-identical, distinguishable particles that each carry a different color. In squaring this form of the amplitude to construct the momentum distribution on phase space, there will be contributions of two general forms that have a different physical interpretation. The squared amplitude can be expressed as:

$$|\mathcal{A}|^2 = \sum_n |\mathbb{T}_n|^2 |A_n|^2 + 2 \sum_{n \neq n'} \text{Re} \left[ \mathbb{T}_n^\dagger \mathbb{T}_{n'} A_n^* A_{n'} \right], \tag{2.2}$$

where the first sum on the right exhibits no interference between different color orderings, and the second sum is the interference due to distinct color orderings. This interpretation strictly only holds at lowest perturbative order at which color interference is manifest, which is what we consider in this paper. Color conservation, that the sum of all color matrices in a gauge-invariant amplitude vanishes, can move products of amplitudes around and correspondingly affect their interpretation. This is especially important to disentangle in a parton shower in which one wants to model color interference effects between numerous soft and/or collinear particles. We will address this again briefly when we present results in a Monte Carlo parton shower, but extending this construction beyond leading order is clearly interesting for future study.

We would like to construct an observable on particle momentum phase space that provides optimal discrimination between these two contributions with the goal of conclusively observing color interference effects. This is therefore a binary discrimination problem and by the Neyman-Pearson lemma [19], the optimal discriminant is the likelihood ratio $\mathcal{L}$, just the ratio of the two contributions:

$$\mathcal{L} \equiv \frac{2 \sum_{n \neq n'} \text{Re} \left[ \mathbb{T}_n^\dagger \mathbb{T}_{n'} A_n^* A_{n'} \right]}{\sum_n |\mathbb{T}_n|^2 |A_n|^2}. \tag{2.3}$$

For an amplitude of a general scattering process, not much more simplification can be achieved. However, for the minimal number of colored particles that exhibit interference, the likelihood ratio can be re-written in a simple, illustrative form.

## 2.1 Color Interference in $e^+ e^- \to q g g \bar{q}$ Events

The minimal number of colored particles necessary to exhibit non-trivial color interference in a scattering amplitude is four. In this section, we consider the process $e^+ e^- \to q g g \bar{q}$ as exemplar of this minimal configuration. First, we note that the processes $e^+ e^- \to q \bar{q}$ and $e^+ e^- \to q g \bar{q}$ exhibit no color interference for a simple reason. The initial electron-positron state is colorless, and so all that is relevant for the color-ordered subamplitudes is the ordering of the final state quarks and gluon (if present). Color-ordered amplitudes are unique up to cyclic permutations and reflections of the order of particles that leave nearest neighbors in the amplitude unchanged. For at most 3 particles, all possible permutations are generated by cyclic permutations and reflections; therefore, there is only one unique color ordering and correspondingly no interference.

On $e^+e^- \to qgg\bar{q}$, there are two distinct color orderings that correspond to interchanging the order of the two gluons. For this process, the likelihood ratio of the interference contribution to the non-interfering terms is

$$\mathcal{L} = \frac{\mathrm{tr}(T_1 T_2 T_1 T_2) \left[ A(q, g_1, g_2, \bar{q})^* A(q, g_2, g_1, \bar{q}) + A(q, g_2, g_1, \bar{q})^* A(q, g_1, g_2, \bar{q}) \right]}{\mathrm{tr}(T_2 T_1 T_1 T_2) \left[ |A(q, g_1, g_2, \bar{q})|^2 + |A(q, g_2, g_1, \bar{q})|^2 \right]} \ . \tag{2.4}$$

In the amplitudes, we have suppressed the initial state $e^+e^-$, and added subscripts to the gluons to distinguish them. $T_i$ is an adjoint color matrix associated with gluon $i$, and we take the trace because the quarks live in the fundamental representation of SU(3) color. For a general SU($N_c$) gauge group, the product of a gluon's color matrix with itself is

$$T_i^j T_k^l = \delta_i^l \delta_k^j - \frac{1}{N_c} \delta_i^j \delta_k^l \ , \tag{2.5}$$

where the indices $i, j, k, l$ represent the rows and columns of the color matrix. With this result, the color matrix traces are

$$\mathrm{tr}(T_1 T_2 T_1 T_2) = -N_c + \frac{1}{N_c} \ , \tag{2.6}$$

$$\mathrm{tr}(T_2 T_1 T_1 T_2) = N_c \left( N_c - \frac{1}{N_c} \right)^2 \ . \tag{2.7}$$

Thus, this likelihood ratio observable $\mathcal{L}$ is explicitly a finite $N_c$ observable. If $N_c \to \infty$, the S-matrix element has no subleading color contributions, and the corresponding distribution will only have support near $\mathcal{L} = 0$.

Any monotonic function of the likelihood ratio is still an optimal discriminant, so we are free to modify the likelihood as we see convenient. As written, the numerator factor of Eq. 2.4 is not a squared amplitude itself and so its physical interpretation is a bit unclear. However, we can freely add to the numerator the squared color-ordered amplitudes in the denominator. Further, the traces over color matrices are just numbers, so we can eliminate them, and still have an optimal discriminant. So, we consider the observable

$$\mathcal{O} \equiv \frac{|A(q, g_1, g_2, \bar{q})|^2 + |A(q, g_2, g_1, \bar{q})|^2 + 2\,\mathrm{Re}\left( A(q, g_1, g_2, \bar{q})^* A(q, g_2, g_1, \bar{q}) \right)}{|A(q, g_1, g_2, \bar{q})|^2 + |A(q, g_2, g_1, \bar{q})|^2} \ , \tag{2.8}$$

that is still maximally sensitive to subleading color effects as it is related to the likelihood $\mathcal{L}$ by a monotonic function, namely:

$$\mathcal{O} = 1 + \frac{\mathrm{tr}(T_2 T_1 T_1 T_2)}{\mathrm{tr}(T_1 T_2 T_1 T_2)} \mathcal{L} \ . \tag{2.9}$$

In this form, the numerator and denominator have a clear interpretation. The numerator is the squared sum of the symmetrized color-ordered amplitudes, and so therefore is just the squared Abelian amplitude:

$$|A(q, g_1, g_2, \bar{q}) + A(q, g_2, g_1, \bar{q})|^2 \equiv |A(q, \gamma, \gamma, \bar{q})|^2 \ . \tag{2.10}$$

Thus, in the numerator, we can treat the massless vector bosons as photons. The denominator is the $N_c \to \infty$ limit of the squared QCD amplitude:

$$|A(q, g_1, g_2, \bar{q})|^2 + |A(q, g_2, g_1, \bar{q})|^2 \equiv |A_{N_c \to \infty}(q, g, g, \bar{q})|^2 . \tag{2.11}$$

Our observable of interest in terms of these amplitudes is then

$$\mathcal{O} = \frac{|A(q, \gamma, \gamma, \bar{q})|^2}{|A_{N_c \to \infty}(q, g, g, \bar{q})|^2} . \tag{2.12}$$

Using the old results for $e^+e^- \to 4$ jets amplitudes of Refs. [30–35], this can be expressed in terms of the momenta of the final state particles. Our goal will be to identify the large-$N_c$ and Abelian contributions to the QCD amplitude to measure and observe color interference.

With this decomposition, it is useful to rewrite the QCD amplitude in terms of the large-$N_c$ amplitude and the QED amplitude. The squared amplitude with full color included can be expressed as

$$|\mathcal{A}(q, g, g, \bar{q})|^2 = [\text{tr}(T_2 T_1 T_1 T_2) - \text{tr}(T_1 T_2 T_1 T_2)] \, |A_{N_c \to \infty}(q, g, g, \bar{q})|^2 + \text{tr}(T_1 T_2 T_1 T_2) \, |A(q, \gamma, \gamma, \bar{q})|^2$$

$$= \left( N_c - \frac{1}{N_c} \right) \left( N_c^2 \, |A_{N_c \to \infty}(q, g, g, \bar{q})|^2 - |A(q, \gamma, \gamma, \bar{q})|^2 \right) . \tag{2.13}$$

This form renders the Abelian amplitude manifestly subleading in $N_c$. Further, in this form, the QCD amplitude has been expressed in terms of a contribution where gluons are completely distinguishable, the large-$N_c$ limit where all gluons have a distinct color, and the Abelian limit, where all gluons are identical particles. In QCD at intermediate $N_c$, gluons are neither perfectly distinguishable nor indistinguishable. This form of the amplitude also invites an interpretation as a sum of a mixed state and a pure state of gluons. We will return to this interpretation in Sec. 2.4.

## 2.2 Color Interference in the Collinear Limit

While the color interference observable constructed from the complete amplitudes can be studied on their own right, they would only be applicable to events from $e^+e^-$ collisions. At a hadron collider, like the LHC, there is never such a pure parton flavor contribution due to non-trivial parton distribution functions. Additionally, because the initial state in the hard scattering at a hadron collider carries color, even $2 \to 2$ processes have non-trivial color interference effects, and any higher final state multiplicity will significantly complicate the structure of the observable. So, to maintain sensitivity to color interference while keeping the observable simple, we will focus on color interference within identified jets. In particular, we formally assume that the radius of the jets $R \ll 1$ and will work to leading order in the collinear limit. In the limit where the gluons become collinear to the quark in the $e^+e^- \to qgg\bar{q}$ process, for example, the squared amplitude factorizes as [36, 37]

$$|\mathcal{A}(q, g, g, \bar{q})|^2 \to |\mathcal{A}(q, \bar{q})|^2 \left( \frac{8\pi\alpha_s}{s_{qgg}} \right)^2 P_{ggq \leftarrow q} . \tag{2.14}$$

Here, $|\mathcal{A}(q,\bar{q})|^2$ is the squared amplitude for $e^+e^- \to q\bar{q}$ events, $s_{qgg}$ is the invariant mass of the final state quark and two gluons, and $P_{ggq \leftarrow q}$ is the universal collinear splitting function. In this limit, all non-trivial color is carried by the splitting function, and so we can exclusively analyze it to establish color interference observables applicable to jets produced in any collider environment.

The splitting function is often expressed as a linear combination of Abelian and non-Abelian contributions, as defined by the color Casimir that multiplies each term:

$$P_{ggq \leftarrow q} = C_F^2 P_{ggq \leftarrow q}^{\mathrm{ab}} + C_F C_A P_{ggq \leftarrow q}^{\mathrm{nab}}. \qquad (2.15)$$

Here, $C_F$ and $C_A$ are the fundamental and adjoint quadratic Casimirs for the $\mathrm{SU}(N_c)$ color group, where

$$C_F = \frac{N_c^2 - 1}{2N_c}, \qquad\qquad C_A = N_c. \qquad (2.16)$$

In QCD, with $N_c = 3$, $C_F = 4/3$, $C_A = 3$. The two color channel splitting functions are referred to as the Abelian ("ab") and non-Abelian ("nab") contributions. The Abelian splitting function is identical to that in QED, and so would be associated with the squared photon emission amplitude from earlier. The non-Abelian splitting function does not have a physical interpretation on its own, but can be combined with the Abelian splitting function to construct the large-$N_c$ splitting function. With the goal of separating the photon contribution from the large-$N_c$ contribution, we can express the splitting function as

$$\begin{aligned} P_{ggq \leftarrow q} &= C_F^2 P_{ggq \leftarrow q}^{\mathrm{ab}} + C_F C_A P_{ggq \leftarrow q}^{\mathrm{nab}} \qquad (2.17) \\ &= \frac{C_F C_A}{2} \left( P_{ggq \leftarrow q}^{\mathrm{ab}} + 2 P_{ggq \leftarrow q}^{\mathrm{nab}} \right) + \frac{C_F}{2}(2C_F - C_A) P_{ggq \leftarrow q}^{\mathrm{ab}}. \end{aligned}$$

The first splitting function, $P_{ggq \leftarrow q}^{\mathrm{ab}} + 2 P_{ggq \leftarrow q}^{\mathrm{nab}}$, is the large-$N_c$ limit, because at large-$N_c$, $C_A = 2C_F$. Note also that the relative size of the large-$N_c$ and Abelian splitting functions is the ratio of color factors

$$\frac{2C_F - C_A}{C_A} = -\frac{1}{N_c^2}, \qquad (2.18)$$

which is exactly the same ratio as established in the form of the squared amplitude from Eq. 2.13.

Then, in the collinear limit, the color interference observable we consider is the ratio of these splitting functions:

$$\mathcal{O} = \frac{P_{ggq \leftarrow q}^{\mathrm{ab}}}{P_{ggq \leftarrow q}^{\mathrm{ab}} + 2 P_{ggq \leftarrow q}^{\mathrm{nab}}}. \qquad (2.19)$$

Explicit expressions for the splitting functions as functions of the energy fractions of the particles in the splitting and their pairwise angles can be found in Refs. [36, 37]. In general, these observables are complicated and unwieldy, and are not easily interpretable from their analytic form. So, we will not present them here. However, the expressions do simplify significantly when the helicities of the gluons are identical, and we will present explicit results for this limited case in the next section.

## 2.3 Explicit Form of Color Interference Observable with Identical Gluon Helicities

When the two gluons have the same helicity, the amplitudes are of the so-called maximally helicity violating (MHV) form, and take a very simple expression. Consider the tree-level color-ordered amplitude for the process $l^+\bar{l}^- \to q^+ g_1^+ g_2^+ \bar{q}^-$:

$$A(q^+, g_1^+, g_2^+, \bar{q}^-) \propto \frac{\langle \bar{q}\bar{l}\rangle^2}{\langle 12\rangle\langle 1q\rangle\langle 2\bar{q}\rangle\langle l\bar{l}\rangle} \,, \tag{2.20}$$

where $l$ ($\bar{l}$) is an initial (anti-)lepton and particle helicities are denoted by the superscripts. We ignore overall coupling factors, focusing on the kinematic dependence exclusively. The angle brackets denote the helicity spinor product, and up to a phase, correspond to the square-root of the invariant mass of the two particles in the product. Standard references on spinor helicity notation are Refs. [24, 38]. The amplitude when the gluons both have negative helicity is related by complex conjugation and interchanging of the quark and anti-quark, and so will produce the same absolute square, so we do not need to consider it explicitly. The square of this amplitude with gluons of positive helicity is

$$|A(q^+, g_1^+, g_2^+, \bar{q}^-)|^2 \propto \frac{s_{\bar{q}\bar{l}}^2}{s_{l\bar{l}}s_{12}s_{1q}s_{2\bar{q}}} \,, \tag{2.21}$$

where $s_{ij}$ is the invariant mass of particles $i$ and $j$. Summing this with the permuted squared amplitude $1 \leftrightarrow 2$ we produce the large-$N_c$ limit:

$$\left|A_{N_c\to\infty}(q^+, g^+, g^+, \bar{q}^-)\right|^2 = \frac{s_{\bar{q}\bar{l}}^2(s_{1q}s_{2\bar{q}} + s_{2q}s_{1\bar{q}})}{s_{l\bar{l}}s_{12}s_{1q}s_{2\bar{q}}s_{2q}s_{1\bar{q}}} \,. \tag{2.22}$$

The Abelian amplitude is found by summing together the color-ordered amplitude with the amplitude in which gluons 1 and 2 are permuted:

$$A(q^+, \gamma^+, \gamma^+, \bar{q}^-) = A(q^+, g_1^+, g_2^+, \bar{q}^-) + A(q^+, g_2^+, g_1^+, \bar{q}^-) \tag{2.23}$$

$$\propto -\frac{\langle \bar{q}\bar{l}\rangle^2\langle q\bar{q}\rangle}{\langle 1q\rangle\langle 2\bar{q}\rangle\langle 2q\rangle\langle 1\bar{q}\rangle\langle l\bar{l}\rangle} \,.$$

Its square is therefore

$$|A(q^+, \gamma^+, \gamma^+, \bar{q}^-)|^2 \propto \frac{s_{\bar{q}\bar{l}}^2 s_{q\bar{q}}}{s_{l\bar{l}}s_{1q}s_{2\bar{q}}s_{2q}s_{1\bar{q}}} \,. \tag{2.24}$$

The ratio of these two squared amplitudes with fixed helicity dramatically simplifies to

$$\frac{|A(q^+, \gamma^+, \gamma^+, \bar{q}^-)|^2}{|A_{N_c\to\infty}(q^+, g^+, g^+, \bar{q}^-)|^2} = \frac{s_{12}s_{q\bar{q}}}{s_{1q}s_{2\bar{q}} + s_{2q}s_{1\bar{q}}} \,. \tag{2.25}$$

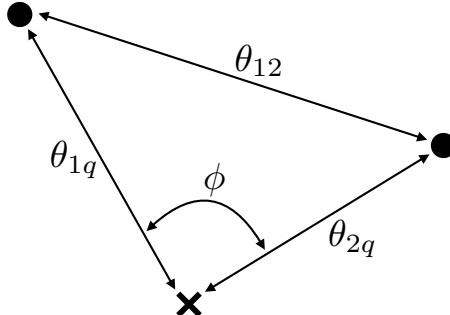

**Figure 1**: Illustration of the configuration of two gluons (dots) about a quark (cross) in the collinear limit on the celestial sphere. Relative angles between pairs of particles and about the direction of the quark are shown.

Further, note that every term in the ratio is homogeneous in the energies of the final state particles, so this observable can be expressed exclusively in terms of particle pairwise angles $\theta_{ij}$:

$$\frac{|A(q^+, \gamma^+, \gamma^+, \bar{q}^-)|^2}{|A_{N_c \to \infty}(q^+, g^+, g^+, \bar{q}^-)|^2} = \frac{(1 - \cos\theta_{12})(1 - \cos\theta_{q\bar{q}})}{(1 - \cos\theta_{1q})(1 - \cos\theta_{2\bar{q}}) + (1 - \cos\theta_{2q})(1 - \cos\theta_{1\bar{q}})} \,. \quad (2.26)$$

In this form, it is trivial to take the collinear limit. With the gluons collinear to the quark, we have $\theta_{12}, \theta_{1q}, \theta_{2q} \to 0$, but no assumed hierarchy between them. Additionally, the anti-quark is antipodal to all other particles, so $\theta_{i\bar{q}} \to \pi$. Then the Abelian to large-$N_c$ ratio reduces to

$$\lim_{q,g,g \text{ collinear}} \frac{|A(q^+, \gamma^+, \gamma^+, \bar{q}^-)|^2}{|A_{N_c \to \infty}(q^+, g^+, g^+, \bar{q}^-)|^2} \to \frac{\theta_{12}^2}{\theta_{1q}^2 + \theta_{2q}^2} \,. \quad (2.27)$$

This is effectively identical in form to the jet color ring introduced in Ref. [25] as an observable for discrimination of color-singlet resonance decays to quarks from massive gluon splitting to quarks. Related observables were established through application of machine learning in Ref. [39].

As this ratio observable is a measure of interference between distinct color states of the gluons, it should manifest as a sinusoidal oscillation. This can be directly observed by re-expressing the ratio of pairwise angles in Eq. (2.27) through the law of cosines. We have

$$\theta_{12}^2 = \theta_{1q}^2 + \theta_{2q}^2 - 2\theta_{1q}\theta_{2q}\cos\phi \,, \quad (2.28)$$

where $\phi$ is the azimuthal angle between the two gluons with respect to the quark. This configuration of particles on the celestial sphere is illustrated in Fig. 1. Then,

$$\lim_{q,g,g \text{ collinear}} \frac{|A(q^+, \gamma^+, \gamma^+, \bar{q}^-)|^2}{|A_{N_c \to \infty}(q^+, g^+, g^+, \bar{q}^-)|^2} \to 1 - \frac{2\theta_{1q}\theta_{2q}}{\theta_{1q}^2 + \theta_{2q}^2}\cos\phi \,, \quad (2.29)$$

clearly illustrating the interference through azimuthal modulation. For fixed angles $\theta_{1q}, \theta_{2q}$ from the quark, if the gluons are close in azimuth, then the value of the observable is less than 1 and correspondingly the large-$N_c$ amplitude is larger than the Abelian amplitude. At large-$N_c$, the gluons would have a collinear singularity with each other. By contrast, if the gluons are on opposite sides of the quark from one another, the Abelian matrix element is larger. Photons are emitted off of the quark independently, and so their distribution would be flat in azimuth.

The form of the ratio of Abelian and large-$N_c$ matrix elements is very special for this helicity configuration; namely all energy dependence of the particles drops out. Including other helicity configurations would spoil this feature, and complicate the interpretation of the observable. Further, a naive application of the color observable as this energy-independent ratio of angles is not IRC safe, and cannot be calculated in fixed-order perturbation theory. This is problematic, because it was through fixed-order perturbation theory that we were able to define this ratio in the first place. We will address and solve these issues in Sec. 3, and provide a simple IRC safe definition of the color interference observable that can be applied generally.

## 2.4 Interpretation of Color Interference and (In)distinguishibility of the Gluon

We have identified color interference through comparison of the large-$N_c$ squared amplitude $|A_{N_c \to \infty}(q, g, g, \bar{q})|^2$ and the Abelian amplitude $|A(q, \gamma, \gamma, \bar{q})|^2$. These two amplitudes are ends of a spectrum, and QCD lies in between them. At truly large-$N_c$, every gluon carries a different color than every other gluon. As such, gluons at large-$N_c$ are distinguishable particles. Distinct configurations of distinguishable particles sum incoherently or at the squared amplitude level, as we observed above. Thus, a large-$N_c$ squared amplitude describes a completely random mixed state of distinguishable gluons, lacking any interference.

By contrast, all photons are identical particles in any process, and so distinct configurations must be summed together coherently, at the amplitude level. A collection of photons is therefore a pure state that exhibits non-trivial interference due to indistinguishability. With, $N_c = 3$, gluons in QCD are neither completely distinguishable nor identical, and so are some intermediate mixed state whose purity is quantified by the relative size of the color factors,

$$\frac{C_A - 2C_F}{C_A} = \frac{1}{N_c^2} \,, \tag{2.30}$$

which is about 10% in QCD.

This can be made more concrete by constructing the density matrices for these pure and mixed states. We will ignore helicity assignments here for compactness, but anyway helicity is observable and so helicity states sum incoherently. The pure state $|\psi\rangle$ of two identical photons is

$$|\psi\rangle = \frac{|\gamma_1 \gamma_2\rangle + |\gamma_2 \gamma_1\rangle}{\sqrt{2}} \,. \tag{2.31}$$

Then, its density matrix in the space spanned by the two photon states is

$$\rho_{\gamma\gamma} = |\psi\rangle\langle\psi| = \frac{1}{2}|\gamma_1\gamma_2\rangle\langle\gamma_1\gamma_2| + \frac{1}{2}|\gamma_1\gamma_2\rangle\langle\gamma_2\gamma_1| + \frac{1}{2}|\gamma_2\gamma_1\rangle\langle\gamma_1\gamma_2| + \frac{1}{2}|\gamma_2\gamma_1\rangle\langle\gamma_2\gamma_1|. \qquad (2.32)$$

As required by conservation of probability, $\mathrm{tr}\,\rho_{\gamma\gamma} = 1$ and as a pure state its density matrix is idempotent: $\rho_{\gamma\gamma}^2 = \rho_{\gamma\gamma}$. On the other hand, the large-$N_c$ two-gluon state is completely random, and so its density matrix is

$$\rho_{gg} = \frac{1}{2}|g_1g_2\rangle\langle g_1g_2| + \frac{1}{2}|g_2g_1\rangle\langle g_2g_1|, \qquad (2.33)$$

for which $\mathrm{tr}\,\rho_{gg} = 1$, but $\rho_{gg}^2 \neq \rho_{gg}$.

In QCD, the two-gluon density matrix would take the form:

$$\rho_{\mathrm{QCD}} = \frac{1}{2}|g_1g_2\rangle\langle g_1g_2| + \frac{1}{2}|g_2g_1\rangle\langle g_2g_1| + \frac{2C_F - C_A}{4C_F}\left(|g_1g_2\rangle\langle g_2g_1| + |g_2g_1\rangle\langle g_1g_2|\right), \qquad (2.34)$$

which is a linear combination of the photon and large-$N_c$ density matrices according to Eq. (2.13) that ensures conservation of probability, $\mathrm{tr}\,\rho_{\mathrm{QCD}} = 1$. To measure the amount of mixture of this state, we can compute the purity from the eigenvalues of the density matrix. The eigenvalues $\lambda_1, \lambda_2$ of the QCD two-gluon density matrix are

$$\lambda_1 = \frac{C_A}{4C_F}, \qquad\qquad\qquad \lambda_2 = 1 - \frac{C_A}{4C_F}. \qquad (2.35)$$

The purity is the trace of the square of the density matrix, $\mathrm{tr}\,\rho_{\mathrm{QCD}}^2$, or

$$\mathrm{tr}\,\rho_{\mathrm{QCD}}^2 = \lambda_1^2 + \lambda_2^2 = 1 - \frac{C_A}{8C_F^2}(4C_F - C_A) = 1 - \frac{N_c^2}{2}\frac{N_c^2 - 2}{(N_c^2 - 1)^2} = \frac{1}{2} + \frac{1}{2N_c^4} + \cdots. \qquad (2.36)$$

On the farthest right equation, we have expanded the purity about the $N_c \to \infty$ limit. Recall that for a completely random ensemble of two states, the purity is $1/2$, thus two gluons in QCD has a density matrix that is very slightly less than completely random. Other measures of purity or entropy can easily be calculated from this QCD density matrix, but the same conclusion would be reached, so we do not study it further here.

Further, two gluons (or any pair of states) can only exhibit quantum interference if their combined state has no non-trivial quantum numbers. Relevant for QCD, this means that two gluons only interfere if their product color state is a singlet. Indeed, the singlet can be formed from multiplying two adjoint gluons as $\mathbf{8} \otimes \mathbf{8} \supset \mathbf{1}$, in SU(3). This interference can also be observed from a color-flow birdtrack diagram [2, 40, 41]. In the collinear limit, we can consider a squared amplitude for two gluon emission off of a quark line and the squared amplitude that would be present in an integral over phase space for calculation of a cross section. A Feynman diagram that represents uncorrelated collinear gluon emission would be, for example, as shown on the left in Fig. 2. The corresponding birdtrack diagram is shown on the right in Fig. 2. Note that there are two closed loops in the birdtrack diagram, which each contribute a factor of $N_c$ to the value of the squared amplitude. Further, as closed

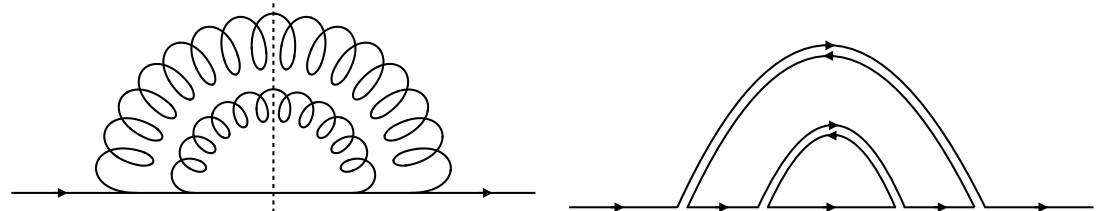

**Figure 2**: Graphical representation of uncorrelated two-gluon emission in the collinear limit from a quark. Left: An example Feynman diagram for the squared amplitude, with the dashed vertical line representing the cut that exposes the final state particles. Right: The corresponding birdtrack diagram that tracks the flow of color amongst the particles.

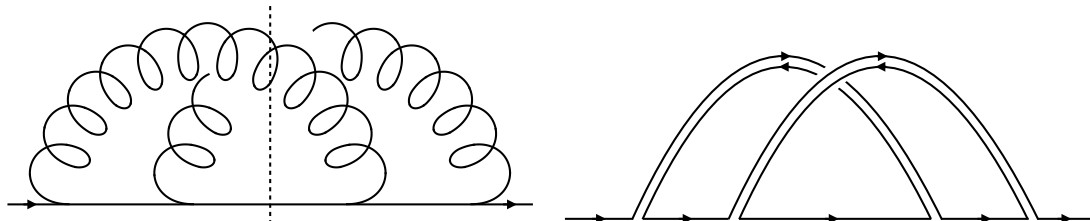

**Figure 3**: Graphical representation of correlated two-gluon emission in the collinear limit from a quark. Left: An example Feynman diagram for the squared amplitude, with the dashed vertical line representing the cut that exposes the final state particles. Right: The corresponding birdtrack diagram that tracks the flow of color amongst the particles.

loops, there is no restriction on their color; the color that flows along each disconnected line is distinct from all others. As such, the colors of the two emitted gluons are not the same and so the gluons are distinguishable, and therefore exhibit no quantum interference through the color quantum number. This is expected because this is a planar birdtrack diagram, and so contributes in the large-$N_c$ limit.

By contrast, we can consider a different squared amplitude, for example that illustrated on the left of Fig. 3, in which the two gluons are emitted in a correlated manner off of the quark. This diagram is non-planar as the gluons that cross the cut have to pass over one another. Its corresponding birdtrack diagram is illustrated on the right of Fig. 3. Unlike the planar birdtrack diagram in Fig. 2, there is only a single color line in this diagram, that just happens to trace out a complicated shape representing the quark and two gluons. Because there is a single color line, this means that the colors of the gluons are necessarily equal, and so they are indistinguishable particles that can and do exhibit quantum interference. Finally, because there are no closed color loops, this diagram is a factor of $1/N_c^2$ suppressed with respect to the planar diagram. Because quantum interference only exists if the gluons have the same color, this can only be guaranteed if there is only one color line in the diagram, and

this is therefore necessarily a subleading color effect.

# 3 Embedding Color Interference in an IRC Safe Observable

The color interference observable we have constructed as a ratio of Abelian to large-$N_c$ matrix elements is sensitive to color interference through the relative angles between two gluons emitted off of a quark. An observable that is exclusively dependent on angles between particles may naively not seem like it is IRC safe, because the lack of energy weighting could mean that arbitrarily soft particles can contribute significantly. This would indeed be the case if we demand that every jet returns a single, unique value associated with the color interference observable. However, if instead we consider energy-weighted cross sections, where contributions to angular distributions are weighted by the energies of the particles being associated, IRC safety can be restored. In this case, each jet will return a distribution of angles, which will be populated according to the dominant energy flow.

To accomplish this, we will describe the orientation of the emitted gluons off of the quark with the three-point energy correlator [26], a generalization of the energy-energy correlator [27] introduced long ago to probe emission or antenna patterns of QCD radiation. We will now work exclusively to leading order in the collinear limit of a jet. Anticipating future experimental applications and a robust way to identify the intiating quark, we will identify the pattern of radiation about a bottom quark. We then define the energy-weighted cross section from the three-point energy correlator relevant for this problem as:

$$\frac{d^3\sigma}{dz_1\,dz_2\,dz_{12}} \equiv \sum_{i,j} \int d\sigma\, \frac{E_b E_i E_j}{Q^3}\, \delta\left(z_1 - \frac{\theta_{bi}^2}{4}\right) \delta\left(z_2 - \frac{\theta_{bj}^2}{4}\right) \delta\left(z_{12} - \frac{\theta_{ij}^2}{4}\right). \tag{3.1}$$

In this expression, the bottom quark is denoted by $b$, and the cross section is differential in three energy-weighted angles: the angles between the bottom quark and either particle $i$ or $j$ in the jet, and the angle between particles $i$ and $j$. The sums run over all particles $i$ and $j$ in the jet which has total energy $Q$. This has been expanded in the collinear limit and is expressed in natural phase space coordinates for jets produced in $e^+e^-$ collisions. For jets produced at a hadron collider, energies and angles would be replaced by momentum transverse to the collision beam and pseudorapidity-azimuth distances, respectively. Note that this expression is a bit different than the more inclusive form of the energy correlators [26, 27] for which there are no identified particles and all energies weightings are summed over all particles. For observation of color interference however, it is vital that the quark be identified so that angles about the quark that are sensitive to color interference are unambiguous.

As observed in Sec. 2.3, the azimuthal angle between the gluons about the quark is sensitive to the effects of color interference. So, we will exchange the pairwise angle $z_{12}$ with the azimuthal angle $\phi$, using the law of cosines, where

$$\cos\phi = \frac{\theta_{\hat{b}i}^2 + \theta_{\hat{b}j}^2 - \theta_{ij}^2}{2\theta_{\hat{b}i}\theta_{\hat{b}j}} = \frac{z_1 + z_2 - z_{12}}{2\sqrt{z_1 z_2}}. \tag{3.2}$$

So, the differential cross section we will consider can be related to the three-point energy correlator simply:

$$\frac{d^3\sigma}{dz_1\, dz_2\, d\phi} = \frac{dz_{12}}{d\phi}\frac{d^3\sigma}{dz_1\, dz_2\, dz_{12}} = 2\sqrt{z_1 z_2}\sin\phi\,\frac{d^3\sigma}{dz_1\, dz_2\, dz_{12}}\,. \tag{3.3}$$

Then, we will take this triple-differential cross section in the angles of the two other particles with respect to the quark and their azimuthal angle as a proxy for the relevant phase space coordinates that define the ratio of cross sections observable, Eq. (2.19).

Then, our procedure for identification of the color interference effects from this three-point correlator is as follows. We calculate the triple-differential cross section on our jets from experimental data or from the full-color splitting function, Eq. (2.15), and we will denote this cross section with a superscript (fc). Color interference is due to effects beyond leading color, and so we want to compare this full color measurement or prediction to the leading color prediction as calculated with the leading-color splitting function contribution of Eq. (2.17). We will denote this triple-differential cross section with superscript (lc). Then, color interference will be imprinted through sinusoidal oscillation in the azimuthal angle $\phi$ through the cross section difference and ratio observable $\mathcal{O}(\phi|z_1, z_2)$

$$\mathcal{O}(\phi|z_1, z_2) \equiv \frac{\frac{d^3\sigma^{(\text{fc})}}{dz_1\, dz_2\, d\phi} - \frac{d^3\sigma^{(\text{lc})}}{dz_1\, dz_2\, d\phi}}{\frac{d^3\sigma^{(\text{lc})}}{dz_1\, dz_2\, d\phi}}\,. \tag{3.4}$$

That is, we define $\mathcal{O}$ as a function of the azimuthal angle, conditioned on fixed angles of the particles with respect to the quark. As the difference between the full- and leading-color cross sections, the numerator exclusively consists of subleading-color effects, reproducing the observable formed from the ratio of Abelian to large-$N_c$ splitting functions in Eq. (2.19), but expressed in IRC safe, reduced phase space coordinates.

With this formulation of the interference observable, it is useful to plot the prediction from the splitting functions directly. First, for the case where we reduce the splitting functions to just include the MHV helicity contributions, recall that the ratio of the matrix elements in the collinear limit takes the form

$$\lim_{q,g,g\text{ collinear}} \frac{|A(q^+, \gamma^+, \gamma^+, \bar{q}^-)|^2}{|A_{N_c\to\infty}(q^+, g^+, g^+, \bar{q}^-)|^2} \to 1 - \frac{2\theta_{1q}\theta_{2q}}{\theta_{1q}^2 + \theta_{2q}^2}\cos\phi = 1 - \frac{2\sqrt{z_1 z_2}}{z_1 + z_2}\cos\phi\,. \tag{3.5}$$

On the right, we have just expressed the result in terms of the phase space coordinates of the three-point energy correlator. When embedded in the observable defined in Eq. (3.4), there is an additional multiplicative factor accounting for the suppression at subleading color:

$$\frac{\frac{d^3\sigma^{(\text{fc,MHV})}}{dz_1\, dz_2\, d\phi} - \frac{d^3\sigma^{(\text{lc,MHV})}}{dz_1\, dz_2\, d\phi}}{\frac{d^3\sigma^{(\text{lc,MHV})}}{dz_1\, dz_2\, d\phi}} = -\frac{1}{N_c^2}\left(1 - \frac{2\sqrt{z_1 z_2}}{z_1 + z_2}\cos\phi\right)\,. \tag{3.6}$$

This simple MHV result will be useful for comparing to the distribution of azimuthal angle $\phi$ from the complete splitting functions.

The three-point energy correlator can be calculated on the $q \to qgg$ splitting function from the expression

$$\frac{d^3\sigma}{dz_1\, dz_2\, dz_{12}} = \int d\Phi_3\, \frac{E_b E_1 E_2}{(E_b + E_1 + E_2)^3}\, P_{q\leftarrow qgg}\, \delta\left(z_1 - \frac{\theta_{bi}^2}{4}\right)\, \delta\left(z_2 - \frac{\theta_{bj}^2}{4}\right)\, \delta\left(z_{12} - \frac{\theta_{ij}^2}{4}\right). \tag{3.7}$$

Here, $d\Phi_3$ is differential three-body collinear phase space whose expression can be found in Refs. [42–44] and $E_1$ and $E_2$ are the energies of the two gluons. While a closed-form, analytic expression for the three-point energy correlator in the triple collinear limit was calculated in Ref. [26] and recently computed for arbitrary angles in $\mathcal{N} = 4$ super-Yang-Mills theory [45], the analytic expressions are extremely unwieldy. For making plots we simply perform numerical integrals of the splitting functions over phase space using the implementation of Vegas [46] from the Cuba libraries [47].

We plot the color interference azimuthal distribution from the complete and MHV splitting functions in Fig. 4. In these plots, we have fixed the ratio of the angles of the gluons to the quark $z_1/z_2$ to values distinguished by color. The dominant description of the azimuthal distribution is clearly accounted for by the simple MHV result of Eq. (3.6). Even with the complete splitting function, the distribution exhibits a single node about its mean, independent of the relative angles of the gluons to the quark. This suggests that dependence of the azimuthal distribution on the ratio $z \equiv z_1/z_2$ factorizes into a simple form:

$$\mathcal{O}(\phi|z) = -\frac{1}{N_c^2}\left(1 + f(z)\sum_{n=1}^{\infty} c_n \cos(n\phi)\right), \tag{3.8}$$

for some function $f(z)$ and constant coefficients $c_n$ in the Fourier series. This assumed form can then be used to extract the Fourier coefficients in data and conclusively demonstrate interference due to subleading color effects.

Throughout this analysis, we have ignored the contribution from $g \to q\bar{q}$ splitting. For jets that include a bottom quark, the collinear splitting function $b \to bq\bar{q}$, for $q = u, d, s, c$, should be included because the flavor of the particles emitted off of the bottom quark cannot be determined. However, the numerical effect of including this splitting function on the azimuthal distribution is very small, and is only a modification of about 5% to the leading-color splitting functions.

# 4 Predictions in Monte Carlo

With this understanding and predictions from the fixed-order collinear splitting functions, we now turn to comparison with parton shower Monte Carlos. We generated $e^+e^- \to b\bar{b}$ events in Pythia 8.306 [48] at a center-of-mass collision energy of 2 TeV. Default settings were used, except turning off hadronization. Events were analyzed in FastJet 3.4.0 [49] and we calculated the three-point energy correlator inclusively about all bottom or anti-bottom quarks in every

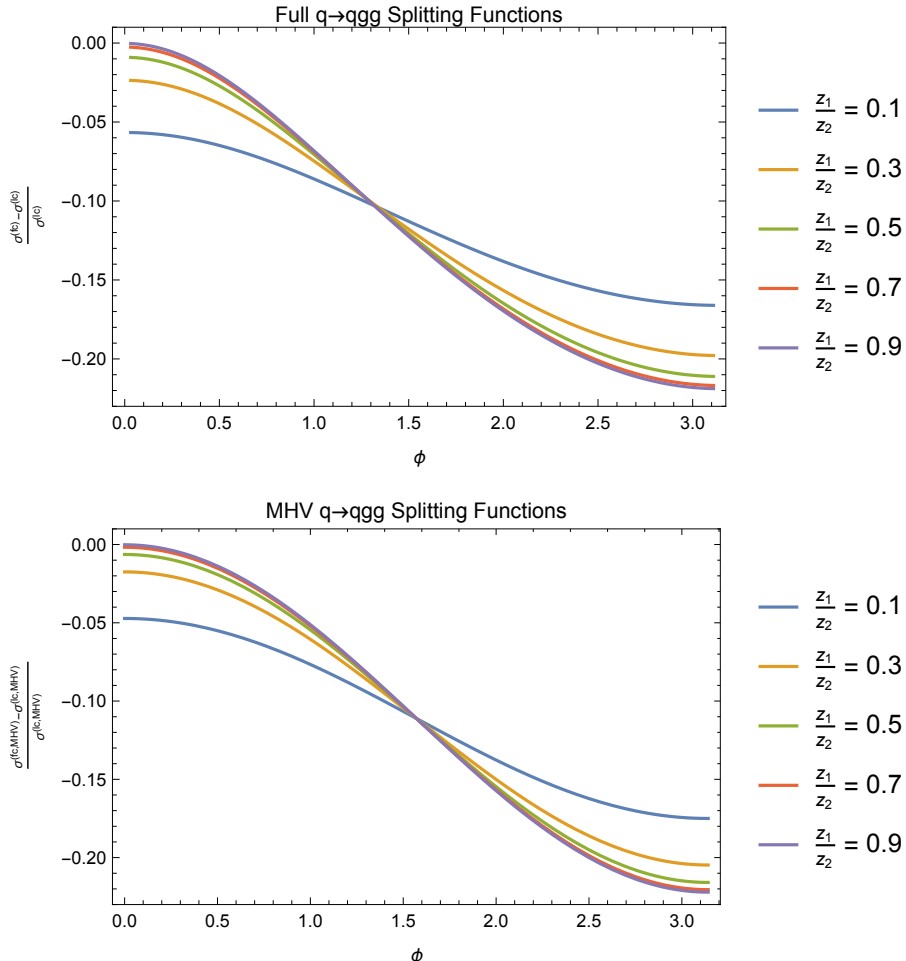

**Figure 4**: Plot of the relative sinusoidal color interference between the full- and leading-color contributions to the complete $q \to qgg$ collinear splitting functions (top) and the MHV collinear splitting function result of Eq. 3.6 (bottom). The colors correspond to different ratios of the distances of the two gluons to the quark.

event. Contributions to the three-point energy correlator were binned by the maximum angle $\theta_{\max}$ of a particle from the bottom quark and we consider $\theta_{\max} = 0.08, 0.04, 0.02$ to ensure that the collinear approximation was valid. Triples of particles contributed to the three-point energy correlator if their $\theta_{\max}$ lay within 5% of one of the values of the bin. Additionally, we binned in the ratio of angles $z_1/z_2$ as studied above, and again, values of the three-point energy correlator contributed if they were within 5% of the specific value of $z_1/z_2$ of the bin. Then, in each bin, we calculated the distribution of the azimuthal angle about the bottom quark in the simulated data. Finally, to study potential subleading color effects, the fixed-order, leading-color distribution of the azimuthal angle was subtracted from the simulated

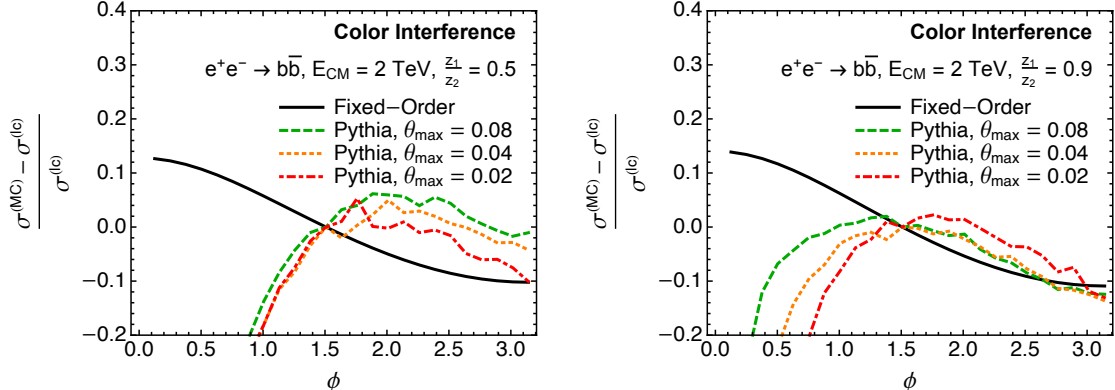

**Figure 5**: Azimuthal angle distribution in the simulated $e^+e^- \to b\bar{b}$ events, scanning over the maximal angle of emissions from the bottom quark, $\theta_{\max}$. Left: ratio of emission angles $z_1/z_2 = 0.5$. Right: ratio of emission angles $z_1/z_2 = 0.9$.

data distribution, and then the leading-color distribution was divided out.

One further point to emphasize for interpretation of these results is that Pythia is a leading-color parton shower in the sense that it does not in general predict subleading color effects correctly even at leading-logarithmic accuracy [13]. However, this does not mean that no subleading color effects are included as angular ordering, a consequence of color coherence, is effectively implemented by the dipole nature of the shower and recoil scheme. Nevertheless, only strict angular-ordered showers implement the proper color factors, see, e.g., Refs. [13, 50]. To address the importance of these included physics and quantify the lack of a complete description of subleading color at even leading-logarithmic accuracy would require either a calculation or simulation that was formally accurate to this order, which we leave to future work.

The results of this procedure are shown in Fig. 5. On the left, we show plots for ratio of emission angles $z_1/z_2 = 0.5$ and on right, $z_1/z_2 = 0.9$. Also plotted is the fixed-order distribution from taking the difference and ratio of the full-color and leading-color distributions, as studied in the previous section. For comparison of the results, we fixed all distributions to vanish near $\phi = \pi/2$. In general, the agreement between the fixed-order expectation and the parton shower simulation isn't great, but we do notice that at large $\phi$ and for large angle ratio $z_1/z_2$, it does appear that there is some modulation of the azimuthal angle, consistent with the effects from subleading color. There are a number of competing effects that are likely responsible for the divergence between the results, especially as $\phi \to 0$.

First, the bottom quark is of course massive, $m_b \simeq 4.2$ GeV, and this mass suppresses collinear emissions below an angle of

$$\theta_{b\ \text{dead}} \simeq \frac{2m_b}{E_b} \gtrsim 0.01\,, \tag{4.1}$$

the so-called dead-cone effect [51–53], where the energy of the bottom quark $E_b$ is less than about half of the center-of-mass energy, 1 TeV. The dead-cone would then seem to be a dominant contribution to the azimuthal distribution for $\theta_{\max} = 0.02$, especially for relatively small angular ratio $z_1/z_2 \lesssim 0.5$. Next, the perturbative parton shower itself terminates at a relative transverse momentum scale on the order of 1 GeV, just above the QCD scale, $\Lambda_{\mathrm{QCD}}$. Then, the minimum angle between particles from the parton shower is

$$\theta_{\min} \simeq \frac{2\Lambda_{\mathrm{QCD}}}{E_b} \gtrsim 0.002 \,. \tag{4.2}$$

This could be a dominant contribution at small azimuthal angle $\phi$, when $z_1/z_2$ is large because collinear gluon splittings would be forbidden by the termination of the parton shower. A final effect, especially at large $z_1/z_2$, arises from all-orders, DGLAP suppression of the energies of the relatively collinear gluons emitted from the bottom quark. At small azimuth and large $z_1/z_2$, there is an additional hierarchy between the maximal splitting angle off of the bottom quark $\theta_l$ and the small relative angle of the two emissions $\theta_s$ that needs to be resummed. At leading-logarithmic accuracy, the size of this suppression is approximately [54]

$$\left( \frac{\alpha_s(\theta_s E_b)}{\alpha_s(\theta_l E_b)} \right)^{\frac{\gamma^{(0)}(3)}{\beta_0}} \gtrsim 0.8 \,. \tag{4.3}$$

The estimate of this 20% suppression assumes that $\theta_s/\theta_l = 0.9$ and $\theta_l = 0.04$, and uses the one-loop coefficient of the QCD $\beta$-function $\beta_0$ and $\gamma^{(0)}(3)$ is a $j = 3$ moment of the timelike splitting functions.

These effects suggest that the most robust phase space region for comparing parton shower simulation to the fixed-order subleading color predictions is for relatively large maximum angle $\theta_{\max}$, relatively large ratio $z_1/z_2$, and at large azimuthal angle $\phi$. Indeed, from Fig. 5, this is the region where there is the best agreement between simulation and analytic prediction, but a more detailed study is needed to concretely establish the robustness of this subleading color effect, especially with realistic hadronic events. Further, to produce these plots, we generate 40 million $e^+e^- \to b\bar{b}$ events and still the statistical fluctuations in the resulting distributions are large (as illustrated by the kinks in the distributions of Fig. 5). This is intrinsically a small effect and is extracted by a subtle subtraction and normalization of factors that are otherwise very similar. However, even with these considerations, the fact that any qualitative agreement is observed whatsoever is encouraging, and motivates further study.

## 5 Conclusions

We have introduced a novel observable that is explicitly sensitive to physics beyond the leading color approximation, due to quantum interference of intermediate states with different color quantum numbers. Our construction of this observable relied on identifying the goal as a binary discrimination problem for which the Neyman-Pearson lemma ensures that the optimal observable is the likelihood ratio. Ratios of distributions on particle phase space are not in

general IRC safe, so we embed this observable into an all-orders IRC safe observable using properties of the multi-point energy correlation functions. For the simplest process exhibiting non-trivial color interference, a collinear jet with three particles, the dominant interference effect is proportional to a $\cos\phi$ modulation, but whose amplitude is suppressed by $1/N_c^2$. Leading-color parton shower Monte Carlos seem to qualitatively agree with the leading color predictions for the distribution of angles of gluons about the initiating quark in the jet.

For this final point, much more analysis is necessary. To firmly establish that leading-color parton shower Monte Carlos cannot describe these effects, we need concrete predictions from parton showers beyond leading color. Many programs exist [4–17], but in general are not available for public use. We urge these collaborations to measure these subleading color observables on the simulated data produced by their parton showers and compare with the full-color results in the collinear limit as defined by the $1 \to 3$ collinear splitting function.

For establishing that a full-color parton shower is accurate at logarithmic accuracy requires further work beyond that presented in this paper. Our predictions have been limited to fixed perturbative order, though in the collinear limit. A parton shower is designed to resum logarithms that arise at all orders in perturbation theory and effects from beyond fixed order may distort the prediction of the sinusoidal interference plotted in Fig. 4. For example, resummation of the collinear logarithms for the interference of intermediate spin states of the gluon in Ref. [1] slightly reduces the amplitude of the interference effect. This has further been established to be described by a next-to-leading logarithmic accurate parton shower [55]. However, it is also important to note that the kinematic regime in which spin interference is manifest is distinct from that where color interference is manifest. First, both are studied in the collinear limit, where all particles in the energy correlator are at small angles to one another. However, spin correlations are further only present when the intermediate gluon goes on-shell, which requires a strong ordering of the collinearity of the daughter particles from the gluon. Thus, there are multiple, hierarchical collinear scales that must be resummed. By contrast, color interference is present when there is no strong ordering of the angles between interfering particles and so its factorization and resummation will be significantly different. We look forward to the development of a framework for analytical resummation in this regime.

Finally, this color interference can in principle be measured in data collected in the experiments at the Large Hadron Collider to establish the existence of quantum interference due to orthogonal intermediate color states. While measurements within the collaborations could firmly establish quantitative comparisons with predictions, less rigorous analyses could be performed on the CMS experiment's data [56, 57] within the CERN OpenData project [58]. Recently, data collected in 2015 was released to the public in which long-live $B$ hadrons have been identified a displaced vertices in the data, e.g., Ref. [59]. The bottom hadron can act as a proxy for the perturbative bottom quark initiating the jet, and therefore defines an axis about which to measure color correlations. This would be an exciting prospect for another avenue of establishing the importance of quantum mechanics to describe collier physics data.

## Acknowledgments

I thank Jack Collins, Kyle Lee, and Ian Moult for collaboration at early stages of this work. A.L. is supported by the Department of Energy, Contract DE-AC02-76SF00515.

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
