# Peer review of "Quantum Interference in Jets at Subleading Color"

_SciPost Physics_

## Round 2 · Referee Report · Anonymous (Referee 1) · 2022-10-9

Report

The author discusses an important issue, namely the question if effects beyond the commonly adopted large-N approximation can actually be discriminated using tailored observables. Especially in light of recent developments which try to extent parton showers and resummation programs beyond the large-N limit, this is an important and timely topic of research. The author also highlights the fact that the large-N limit is not an approximation which relies on "one in nine being a small number", but rather on neglecting interference terms, something which should be communicated and explained in detail much more often. I therefore find the paper worth publishing, but I ask the author to include several remarks which clarify that this study can only be a starting point (possibly also by choosing a less bold title): this has to do with the fact that the paper solely addresses fixed-order examples and entirely neglects the recent development of understanding the all-order structure at subleading colour, which would be the right framework to place the investigation in. Some of the arguments I find a bit misleading, in particular the starting point of diagonal versus off-diagonal colour operators in squaring the amplitude. This is particularly problematic since soft enhanced terms in a physical gauge do cross-talk in between diagonal and off-diagonal colour correlators, and the advocated picture only emerges after exploiting colour conservation. This cross-talk is extremely important when one tries to combine (multiple) collinear and soft(-collinear) limits, an effect which is clearly of utmost importance to disentangle subleading-colour from leading colour and other effects. When considering Monte Carlo predictions one should also take into account that several of the subleading N effects have been included despite their apparent leading colour structure, mainly through coherent branching or emulations thereof. After these issues have been commented on I am happy to fully support the present work for publication. Last not least I should apologize to the author for the delay in delivering my report, which was due to several issues, but also the interest which the present work has sparked with me.
  • validity: -
  • significance: -
  • originality: -
  • clarity: -
  • formatting: -
  • grammar: -

Author:  Andrew Larkoski  on 2022-10-10  [id 2907]

(in reply to Report 1 on 2022-10-09)
Category:
correction

I thank the referee for their interest in this paper and comments on clarification of its presentation.

Following the referee's suggestion, I have retitled the paper "Designing Observables for Quantum Interference in Jets at Subleading Color".

To the end of the first paragraph of section 2, I have added the sentences: "This interpretation strictly only holds at lowest perturbative order at which color interference is manifest, which is what we consider in this paper. Color conservation, that the sum of all color matrices in a gauge-invariant amplitude vanishes, can move products of amplitudes around and correspondingly affect their interpretation. This is especially important to disentangle in a parton shower in which one wants to model color interference effects between numerous soft and/or collinear particles. We will address this again briefly when we present results in a Monte Carlo parton shower, but extending this construction beyond leading order is clearly interesting for future study."

In section 4, I have added a new second paragraph that points out that even LC parton showers contain some subleading effects: "One further point to emphasize for interpretation of these results is that Pythia is a leading-color parton shower in the sense that it does not in general predict subleading color effects correctly even at leading-logarithmic accuracy [13]. However, this does not mean that no subleading color effects are included as angular ordering, a consequence of color coherence, is effectively implemented by the dipole nature of the shower and recoil scheme. To address the importance of these included physics and quantify the lack of a complete description of subleading color at even leading-logarithmic accuracy would require either a calculation or simulation that was formally accurate to this order, which we leave to future work."

I hope that with these edits the paper is acceptable for publication.

Thank you,
Andrew Larkoski

Author:  Andrew Larkoski  on 2022-10-10  [id 2906]

(in reply to Report 1 on 2022-10-09)
Category:
correction

I thank the referee for their interest in this paper and comments on clarification of its presentation.

Following the referee's suggestion, I have retitled the paper "Designing Observables for Quantum Interference in Jets at Subleading Color".

To the end of the first paragraph of section 2, I have added the sentences: "This interpretation strictly only holds at lowest perturbative order at which color interference is manifest, which is what we consider in this paper. Color conservation, that the sum of all color matrices in a gauge-invariant amplitude vanishes, can move products of amplitudes around and correspondingly affect their interpretation. This is especially important to disentangle in a parton shower in which one wants to model color interference effects between numerous soft and/or collinear particles. We will address this again briefly when we present results in a Monte Carlo parton shower, but extending this construction beyond leading order is clearly interesting for future study."

In section 4, I have added a new second paragraph that points out that even LC parton showers contain some subleading effects: "One further point to emphasize for interpretation of these results is that Pythia is a leading-color parton shower in the sense that it does not in general predict subleading color effects correctly even at leading-logarithmic accuracy [13]. However, this does not mean that no subleading color effects are included as angular ordering, a consequence of color coherence, is effectively implemented by the dipole nature of the shower and recoil scheme. To address the importance of these included physics and quantify the lack of a complete description of subleading color at even leading-logarithmic accuracy would require either a calculation or simulation that was formally accurate to this order, which we leave to future work."

I hope that with these edits the paper is acceptable for publication.

Thank you,
Andrew Larkoski

---

## Editorial Decision

resubmitted